# An evolutionarily conserved pathway mediated by neuroparsin-A regulates reproductive plasticity in ants

Xiafang Zhang[1,2], Nianxia Xie[3,4], Guo Ding[5], Dongdong Ning[6], Wei Dai[4], Zijun Xiong[4], Wenjiang Zhong[1,2], Dashuang Zuo[1,2], Jie Zhao[1], Pei Zhang[4,7], Chengyuan Liu[5], Qiye Li[4,7], Hao Ran[1], Weiwei Liu [1] *, Guojie Zhang[5,8] *

1 Key Laboratory of Genetic Evolution & Animal Models, Kunming Institute of Zoology, Chinese Academy of Sciences, Kunming, China, 2 Kunming College of Life Science, University of Chinese Academy of Sciences, Kunming, China, 3 College of Life Sciences, University of Chinese Academy of Sciences, Beijing, China, 4 BGI Research, Wuhan, China, 5 Center of Evolutionary & Organismal Biology, and Women's Hospital at Zhejiang University School of Medicine, Hangzhou, China, 6 College of Agriculture and Biotechnology, Institute of Insect Science, Zhejiang University, Hangzhou, China, 7 State Key Laboratory of Agricultural Genomics, BGI Research, Shenzhen, China, 8 Liangzhu Laboratory, Zhejiang University Medical Center, Hangzhou, China

* liuweiwei@mail.kiz.ac.cn (WL); guojiezhang@zju.edu.cn (GZ)

**Data Availability Statement:** All relevant data are within the paper and its Supporting Information files.

## Abstract

Phenotypic plasticity displayed by an animal in response to different environmental conditions is supposedly crucial for its survival and reproduction. The female adults of some ant lineages display phenotypic plasticity related to reproductive role. In pharaoh ant queens, insemination induces substantial physiological/behavioral changes and implicates remarkable gene regulatory network (GRN) shift in the brain. Here, we report a neuropeptide *neuroparsin A* (*NPA*) showing a conserved expression pattern associated with reproductive activity across ant species. Knock-down of *NPA* in unmated queen enhances ovary activity, whereas injection of NPA peptide in fertilized queen suppresses ovary activity. We found that NPA mainly affected the downstream gene *JHBP* in the ovary, which is positively regulated by NPA and suppression of which induces elevated ovary activity, and *shadow* which is negatively regulated by NPA. Furthermore, we show that NPA was also employed into the brain–ovary axis in regulating the worker reproductive changes in other distantly related species, such as *Harpegnathos venator* ants.

## Introduction

A hallmark of social insects is that different morphology, physiology, and behavior can develop from individuals with a shared genome in response to varying environmental conditions, such as nutrition or social signals [1]. The lifecycle for the majority of ant species begins from insemination of a virgin queen (gyne). After insemination, a queen acquires sperm storage for lifetime usage and begins to be actively engaged in reproduction. The inseminated queen has remarkable extended lifespan compared to un-inseminated gyne, breaking the trade-off

**Funding:** This work was supported by the National Natural Science Foundation of China (grant No. 31900399 and No. 32170631 to W.L., No. 31970573 to G.Z. and No. 32388102 to G.Z) and the New Cornerstone Science Foundation through the XPLORER PRIZE to G.Z. The funders had no role in study design, data collection and analysis, decision to publish, or preparation of the manuscript.

**Competing interests:** The authors have declared that no competing interests.

**Abbreviations:** BSA, bovine serum albumin; CA, corpora allatum; DEG, differentially expressed gene; dsRNA, double-stranded RNA; GRN, gene regulatory network; HCR, hybridization chain reaction; HPG, hypothalamic–pituitary–gonadal; IGFBP, insulin-like growth factor-binding protein; MAPK, mitogen-activated protein kinase; MNSC, medial neurosecretory cell; NPA, neuroparsin A; PBS, phosphate-buffered saline; PI, pars intercerebralis; TPM, transcripts per million.

relationship between longevity and fecundity which is widely observed in solitary insects [2–4]. Insemination is a once-in-a-lifetime event for female reproductive ants and only those successfully mated would become fully functional queens [5]. While in some ant species, such as pharaoh ant *Monomorium pharaonis* and leaf-cutting ant *Acromyrmex echinatior*, gynes failed to mate can stay with the colony permanently and display some worker-like behaviors, such as more explorative locomotion and foraging out of nest, thereby, gaining some indirect fitness by assisting their mother or reproductive sisters to produce more offspring [6,7]. The different physiological and behavioral changes during the gyne/queen transition provide an excellent example of phenotypic plasticity that is widely observed in diversified ant lineages. Interestingly, similar reproductive role transition also occurred in the worker caste of some ant species where some of their workers have maintained reproductive potential. For example, in *Hapegnathos saltator* ant colony, when the queen is absent, workers would compete for the dominant status and the final winner would become reproductively activated as a gamergate (pseudo-queen) [8]. In clonal raider ant *Ooceraea biroi*, queen caste is permanently lost and the workers can reproduce by thelytokous parthenogenesis, cycling between an egg-laying reproductive phase and brood-caring nonreproductive phase [9].

The reproduction of female insects is centrally governed by the brain–ovary axis, which implicates the complex crosstalk between a repertoire of neuropeptide, nutritional and hormone-related components, such as insulin, TOR, juvenile hormone, ecdysone, and vitellogenin associated pathways [10]. Analogous to human HPG (hypothalamic–pituitary–gonadal) axis, the brain–ovary axis of insects could orchestrate a spectrum of physiological and behavioral changes with bidirectional signalings between organs [11–16]. The reproductive groundplan hypothesis proposes that the co-option of the regulatory components for female reproduction may underlie social evolution in insects [17,18], and the central prediction of the hypothesis is that the reproductive physiology and social behaviors are coupled by a pleiotropic genetic network [19,20]. In agreement with this hypothesis, some previous empirical evidences have demonstrated such correlation between ovarian activity and social task performances [21–23]. The reproductive role transition in ant entails coordinated physiological and behavioral changes; however, the molecular mechanism of brain–ovary axis involved in regulating the reproductive and social role difference during these transitions remains to be elucidated.

Previous comparative transcriptomics analysis has identified a repertoire of differentially expressed genes (DEGs) underlying gyne/queen transition, and most of these genes display similar directional expressions in different reproductive roles in the distantly related ant lineages with reproductive workers, suggesting that the same gene regulatory network (GRN) was retained to regulate different forms of reproductive role differentiations across ant species [6]. Among these, *neuroparsin A* (*NPA*) is one of the top ranked and earliest responsive DEGs for gyne/queen transition (suppressed in 5-day-old queen who is inseminated for 2 days), and its expression is suppressed in queen long after insemination (suppressed in 30-day-old queen who is inseminated for 27 days) [6]. *NPA* is found to be exclusively expressed in the medial neurosecretory cells (MNSC) in the brain of pharaoh ants [6], suggesting its potential role as a circulating neuro-hormone that targets remote organs. While *NPA* is absent from the genomes of several *Drosophila* species, including *Drosophila melanogaster* [24], it is found in a wide range of arthropods. Neuroparsin belongs to a family of conserved arthropod neuropeptides, sharing pronounced similarity to vertebrate insulin-like growth factor-binding proteins (IGFBPs), regulating or being regulated by insulin and endocrine hormone signaling pathways [25–31]. It plays pleiotropic functions, such as antidiuretic, increasing lipid and trehalose levels in the circulation, antagonizing juvenile hormone, and interacting with ecdysone for neurotrophic function [27,32–35]. It has been reported to be involved in regulating reproduction in a wide array of arthropods [26]. In primitive eusocial bumble bees, presence of queen induces

increased *neuroparsin* level in worker brain [36]. Previous studies have shown that *Neuroparsin* was consistently found to be repressed when female ants become reproductively activated, including scenarios of gyne/queen transition in *M. pharaonis*, worker/gamergate transition in *H. saltator*, and nonreproductive/reproductive transition in *O. biroi* [6,37,38].

Although comparative transcriptomics studies have revealed the correlative relationship between *NPA* level and reproductive plasticity in ants, the mechanistic investigation of NPA function is scarce and restricted to solitary insects, such as the locusts [25,27,35,39,40]. In this study, we set up distinct reproductive role differentiation models to interrogate whether and how NPA is causally involved in regulating reproductive capability and related behaviors; furthermore, we surveyed whether NPA plays a conserved role in different models of reproductive role differentiations across ant phylogenetic tree.

## Results

### NPA suppression is correlated with reproductive activation across ant phylogenetic tree

To evaluate whether NPA is consistently suppressed in individuals where reproductive function is activated across ant species, we measured *NPA* expression levels between reproductive versus nonreproductive individuals in 9 ant species from 5 different subfamilies. The 9 ant species cover 3 distinct models of reproductive role differentiation: (i) gyne/queen transition as exhibited by *M. pharaonis* and *Tetramorium bicarinatum* from Myrmicinae subfamily, and *Camponotus japonicus*, *Camponotus nicobarensis*, *Oecophylla smaragdina*, *Lasius alienus* from Formicinae subfamily; (ii) worker/gamergate transition as exhibited by *Harpegnathos venator* from Ponerinae subfamily and *Gnamptogenys bicolor* from Ectatomminae subfamily; and (iii) nonreproductive/reproductive worker transition within worker caste as exhibited by *O. biroi* (Fig 1A) [6,41,42].

Among the 6 species with gyne/queen transition model, the *NPA* expression levels in the brain is always down-regulated in queens compared to gynes (Fig 1B). Worker/gamergate transition only appeared in some ant lineages, where workers compete for dominant status by mutilation, aggression, or ritualized behaviors when the queens are lost and the winner workers could replace queen function as gamergates [41,43,44]. In both species with this model, *NPA* expressions were down-regulated in gamergates compared to workers (Fig 1B). Nonreproductive/reproductive worker transition is displayed in a few ant lineages such as *O. biroi* ants which have secondarily become queenless and the workers reproduce by thelytokous parthenogenesis [45,46]. The workers synchronize their activities and alternate between a collective egg-laying phase and a collaborative brood-caring phase [9]. The *NPA* expression was down-regulated in the reproductively activated egg-laying phase compared to the brood-caring phase (Fig 1B). Therefore, the *NPA* suppression is always correlated with the reproductive activation in ants. The correlation is persistently observed across ant phylogenetic tree and maintained in the 3 major reproductive role differentiation models.

In ants, the MNSCs in the brain are the cells responsible for secreting insulin-like peptides, which have been proved to regulate reproduction and lifespan in multiple ant species [4,47]. *NPA* was found to be expressed in the MNSCs in pharaoh ants [6], to address whether the specific expression pattern is also conserved in other ant species, we performed hybridization chain reaction (HCR) experiments to detect the *NPA* transcripts in situ. In all 3 species we examined, including *M. pharaonis*, *H. venator*, and *O. biroi*, we found that *NPA* was exclusively expressed in the MNSCs in the brain (Fig 1C). Furthermore, we found that *NPA* expression in *M. pharaonis* was co-localized with *insulin-like peptide 2* (*Ilp2*, *LOC 105832901*), homolog of which in *O. biroi* was found to be expressed in the same cells and responsible for promoting

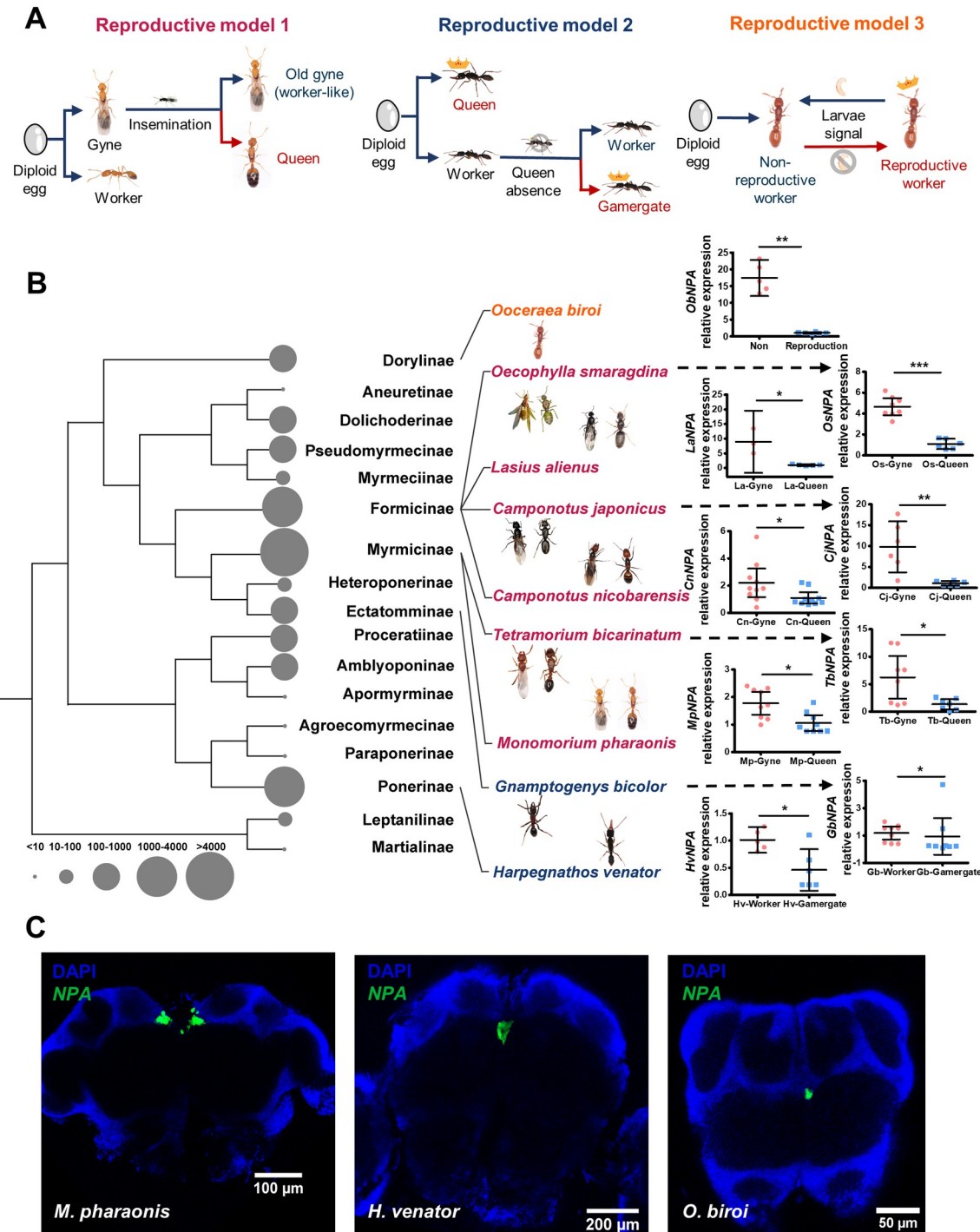

**Fig 1. Neuro-secreted NPA is always suppressed when reproduction is activated across ant species.** (**A**) The 3 representative ant reproductive role differentiation models. Models 1, 2, and 3 indicate gyne/queen transition, worker/gamergate transition, and reproductive/nonreproductive worker transition models, respectively. (**B**) Real-time qPCR experiments quantify *NPA* expression levels of different reproductive roles across ant species (Ob: *O. biroi*, Os: *O. smaragdina*, La: *L. alienus*, Cj: *C. japonicus*, Cn: *C. nicobarensis*, Tb: *T. bicarinatum*, Mp: *M. pharaonis*, Gb: *G. bicolor*, and Hv: *H. venator*). Bars represent mean and 95% CI (Mann–Whitney U test, two-tailed, *$P < 0.05$, **$P < 0.01$, ***$P < 0.001$). The species names were colored in accordance with different reproductive models in panel A. The ant phylogenetic tree was constructed using Sanger-sequencing data sets of 11 nuclear genes, referring to a previous study [48]. The sizes of the circles indicate the estimated species number referring to AntCat (https://antcat.org/). (**C**) In situ HCR detection of *NPA* expression in MNSCs across 3 species, which were representative of reproductive models 1, 2, and 3, respectively. The numerical data for this figure can be found in S1 Data. HCR, hybridization chain reaction; MNSC, medial neurosecretory cell; NPA, neuroparsin A.

ant reproduction (S1 Fig). We aligned NPA and its homologs in 10 species and found 12 to 16 conserved cysteines, which are crucial residues in IGFBP for binding insulin (S1 Fig). The expression pattern of NPA and its homology to IGFBP suggest that NPA acts as a neurosecretory peptide which is released into circulation and can remotely affect target tissues.

## NPA plays anti-gonadotropic function in *M. pharaonis*

To address whether NPA plays a causal role in regulating reproduction, we knock-down *NPA* expression in pharaoh ant gynes by injection of double-stranded RNA (dsRNA) and then examine how ovary activation was affected. We used young gynes as our experimental animals since they have a very low level of vitellogenesis activity before insemination. We collected newly eclosed gynes and injected *NPA* dsRNA (*ds-NPA*) on day 4, day 6, and day 8, while as control groups, we injected *eGFP* dsRNA (*ds-eGFP*) and *NPA*-random dsRNA (*ds-random*) following the same protocol with comparable amount of dsRNA (Fig 2A). On day 9, before the gynes spontaneously lay the first batch of eggs, the *ds-NPA* treated gynes had more yolky oocytes and the sizes of the yolky oocytes were significantly enlarged compared to the control group, indicating that high expression of *NPA* inhibits reproductive activation in gynes, and its down-regulation induced by insemination is necessary for reproductive activation in queens (Fig 2B and 2C).

We further carried out gain-of-function studies to confirm the anti-gonadotropic function of NPA. Since NPA is a neuropeptide which is released into hemolymph and then transported by circulation to target tissues, thereby, direct injection of recombined NPA peptide into hemolymph could elevate its level globally. We choose newly inseminated queens who normally display higher levels of ovary activity while before laying the first batch of eggs as our experimental animals. To specify the date of insemination and the age of the queen, we collected newly eclosed gynes into new rearing boxes on day 0, and introduced over-numbered males (gyne: male≈1:1.5) into the boxes for mating on day 3. On day 4, we removed the males from the boxes, then injected NPA peptide into queens on day 5 and day 7. As control groups, we injected bovine serum albumin (BSA) and heat-inactivated NPA peptide (inactivated-NPA) into queens at the same age following the same protocol with a comparable amount of proteins. On day 9, we dissected out the ovaries of queens and measured their ovary activity levels (Fig 2D). We found that the increase of NPA peptide level caused suppression of ovary activity. Compared to the control groups, active NPA peptide injected queens had significantly fewer number of yolky oocytes and the size of the yolky oocytes was also significantly suppressed (Fig 2E and 2F).

The loss-of-function and gain-of-function studies consistently support the conclusion that NPA is an anti-gonadotropic factor in ants. Its suppression is necessary to trigger an accelerated ovary activation in queens while the higher levels of NPA in gynes suppresses their ovary activation, thereby, the gynes can only display a very low level of spontaneous ovary activity.

## NPA regulates queen-like/worker-like behavioral change during gyne/queen transition

Besides ovary activation, gyne/queen transition also entails a repertoire of behavioral shifts. After insemination, queens would quickly detach their wings, recede into the deep nest, and are fully occupied by egg-laying activity [2]. In contrast, old gynes without insemination would display more worker-like behaviors, including more explorative locomotion out of the nest and notably elevated foraging runs [6]. *NPA* is one of the earliest down-regulated genes in response to insemination and its expression maintains at low level as queen become older, while as un-inseminated gynes grow older, *NPA* expression level can become even higher;

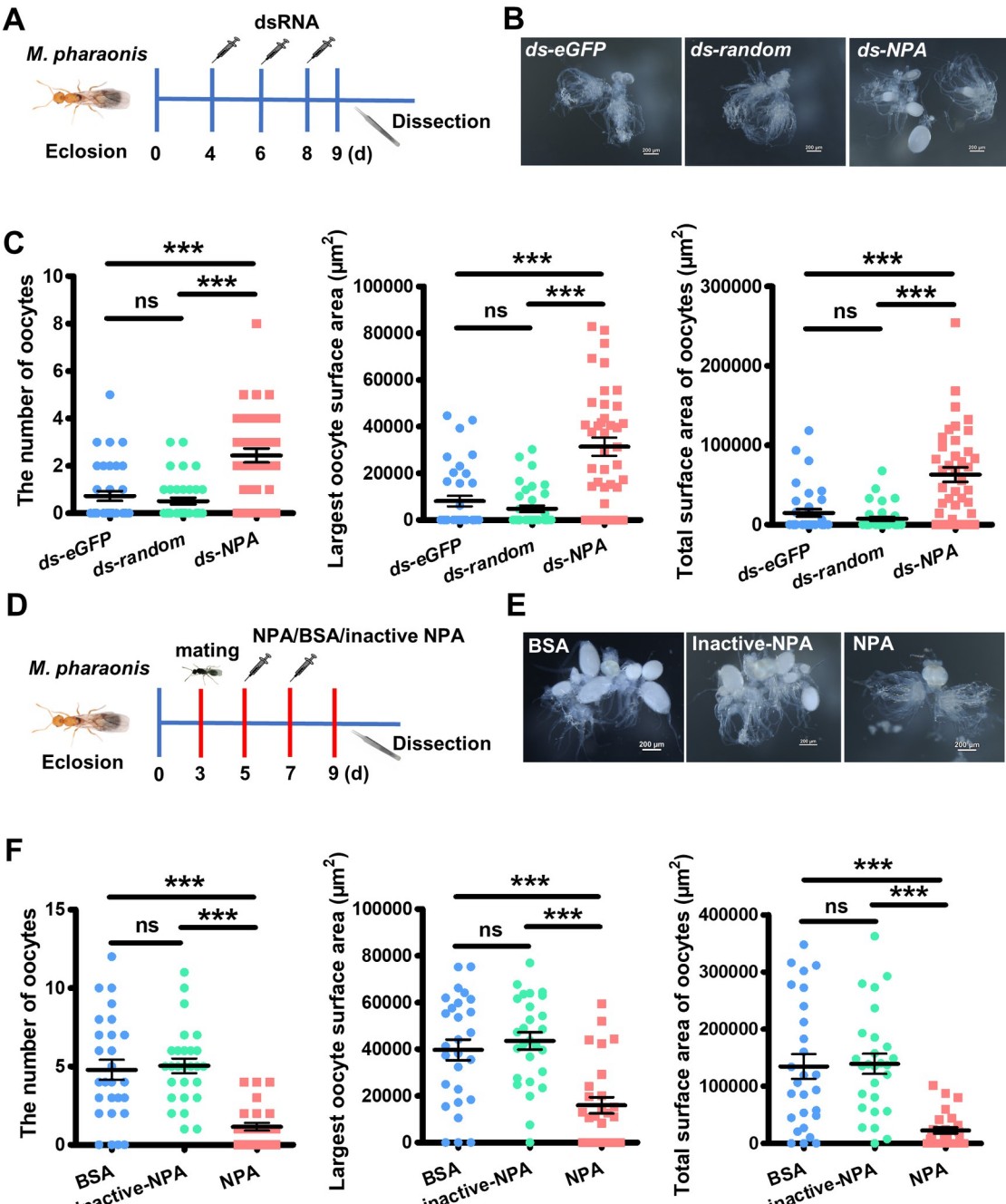

**Fig 2. NPA suppress reproduction in *M. pharaonis*.** (**A**) Schematic diagram of dsRNA injection for reproduction assay. (**B**) Representative ovaries of dsRNA injected gynes, indicating elevated ovary activity after *NPA* knockdown. (**C**) Scatter dot plot shown the oocyte number, the largest oocyte surface area, and the total surface area of oocytes of dsRNA injected gynes. Compared to control groups, *ds-NPA* groups shown elevated ovary activities. Bars represent mean and SEM. ($n = 37$ for *ds-eGFP* group, $n = 37$ for *ds-random* group, and $n = 39$ for *ds-NPA* group, Mann–Whitney U test, two-tailed, ***$P < 0.001$). (**D**) Schematic diagram of NPA peptide, BSA and heated inactive NPA peptide injection for reproductive assay. (**E**) Representative ovaries of NPA peptide, BSA, and heat-inactivated NPA peptide-injected queens, indicating reduced ovary activity after NPA peptide injection. (**F**) Scatter dot plot shown the oocyte number, the largest oocyte surface area, and the total surface area of oocytes of NPA peptide, BSA, and inactive NPA peptide-injected queens. Compared to control groups, NPA peptide-injected groups shown reduced ovary activity. Bars represent mean and SEM ($n = 27$ for BSA-injected group, $n = 27$ for inactive NPA peptide, and $n = 27$ for NPA peptide-injected group, Mann–Whitney U test, two-tailed, ***$P < 0.001$). The numerical data for this figure can be found in S1 Data. BSA, bovine serum albumin; dsRNA, double-stranded RNA; NPA, neuroparsin A.

therefore, it is possible that suppression of *NPA* is not only responsible for triggering the acute ovarian activity but also responsible for sustaining the stereotyped queen behavioral spectrum [6].

To address whether consistently low *NPA* level is necessary for stabilizing queen behaviors, we conducted long-term *NPA* knockdown in *M. pharaonis* by injecting *ds-NPA* into hemolymph of gynes every 4 days for 7 times in total and injected *ds-eGFP* following the same procedure as control group, until the gynes become 30 days old. We recorded and assayed their behaviors during the experiment (S2 Fig); 7 to 9 gynes from either experimental group or control group were introduced into a 9 cm × 9 cm Petri dish and their locomotion behaviors were videotaped. We tracked each individual for its locomotion trajectory, calculated the distance traveled, and generated the heatmap of the visiting frequencies. We found that *ds-NPA*-treated gynes displayed lower visiting frequencies in the heatmap (Fig 3A) and traveled significantly shorter distances than the control group (Fig 3B). We let the gynes habituate to the environment for overnight, then introduced a patch of food into the Petri dish and counted the foraging behaviors. We found that *ds-NPA*-treated gynes exhibited significantly lower foraging runs and shortened foraging durations compared to *ds-eGFP*-treated gynes (Fig 3C).

Conversely, we also increased NPA level for long-term to see if consistently increased NPA could override queen behavioral stereotype and induce some worker-like behaviors. We collected newly eclosed gynes and let them mate on day 3. Then from day 6, we performed NPA peptide injection every 4 days for 6 times in total. As a control group, we injected a comparable amount of BSA following the same procedure. We assayed the behaviors of queens until they were 30 days old (S2 Fig). We found that long-term NPA injection could induce queens to display higher locomotion. The NPA-treated queens displayed higher visiting frequencies in the heatmap (Fig 3D) and traveled significantly longer distances than the BSA-treated queens (Fig 3E). We also detected occasional foraging runs in NPA-treated queens while BSA-treated queens did not show any foraging behavior (S2 Fig).

From the above results, we conclude that consistent NPA suppression is necessary for stabilizing queen stereotyped behaviors. Suppressing NPA levels in gynes could induce them to display queen-like behaviors while artificial elevating NPA in queens could induce them to display worker-like locomotion and foraging behaviors. Thus, NPA is not only an anti-gonadotropic factor for inhibiting ovary activation in gynes, but also it is responsible for promoting worker-like behaviors as gynes become older. NPA suppression is required for mediating the physiological and behavioral shift to true queen functionality.

## NPA regulates worker-gamergate transition in *H. venator*

To confirm whether NPA plays a conserved role in orchestrating the physiological and behavioral shifts underlying reproductive role differentiation across ant species, we used *H. venator* ants to perform *NPA* loss-of-function studies. *H. venator* workers respond to queen absence by dueling responses and the winner becomes the gamergate who can re-engage in reproduction. We removed queens from *H. venator* colonies, then injected *ds-HvNPA* into the hemolymph of workers on day 1, 3, 5, and 7. As a control group, we injected a comparable amount of *ds-eGFP* following the same procedure. On day 9, we measured the ovary activity levels (Fig 4A) and found that compared to *ds-eGFP*-injected workers, *ds-NPA*-injected workers had significantly more yolky oocytes and sizes of the yolky oocytes were also notably enlarged, indicating that *NPA* suppression could induce more ovary activation in *H. venator* workers (Fig 4B and 4C).

To address whether long-term *NPA* suppression could lead to gamergate-like behavioral shift, we conducted *ds-NPA* injection every 4 days for 7 times in total. At 30 days after the first treatment, we assayed the hunting behaviors of the *H. venator* ants (Fig 4D). In the hunting

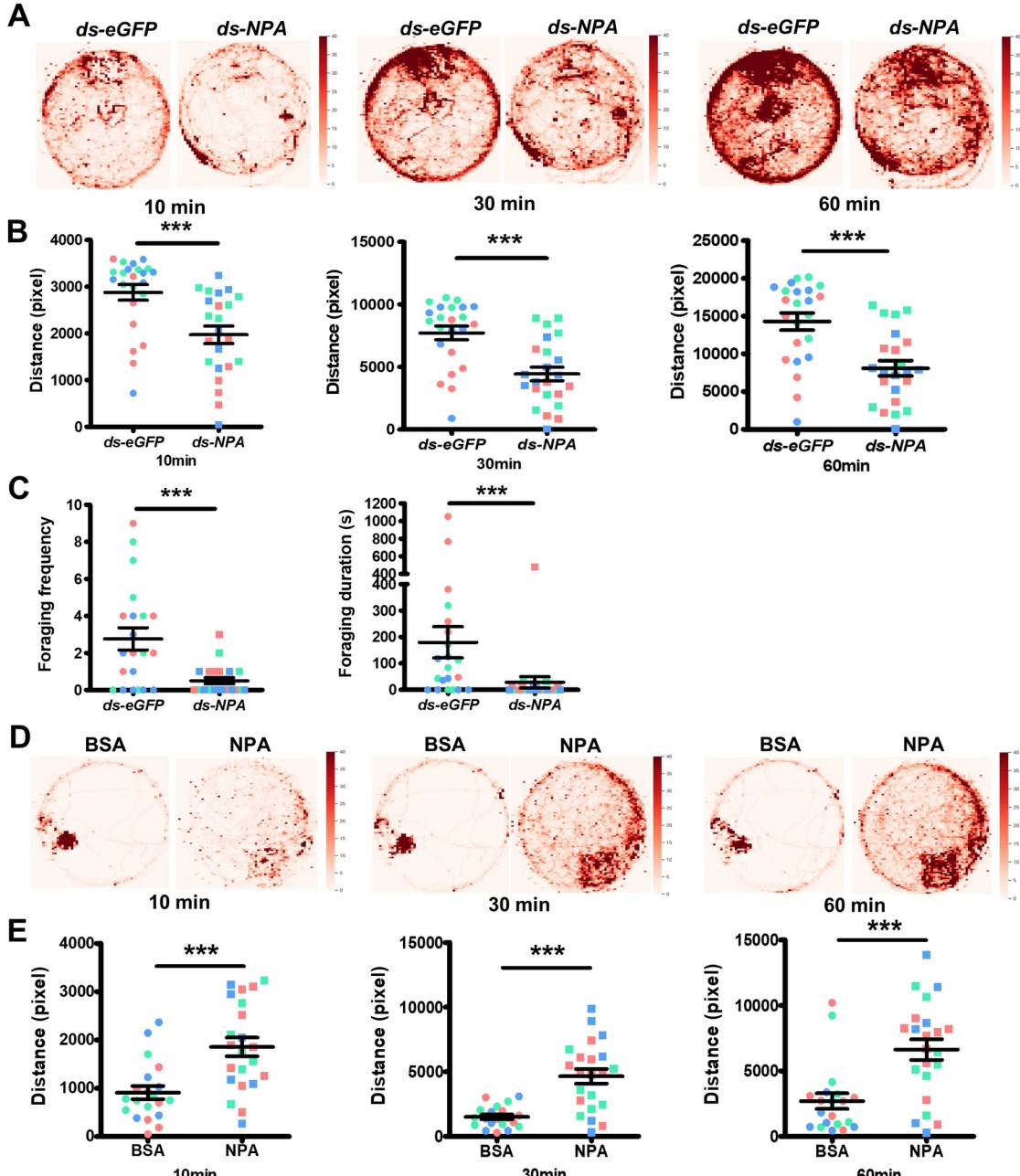

**Fig 3. NPA regulates behavioral shifts for gyne/queen transition in *M. pharaonis*.** (**A**) The locomotion activity after long-term dsRNA injection. Representative heatmaps shown locomotion trajectories of individuals in each group for 10 min, 30 min, and 60 min, indicating reduced locomotion activity after *NPA* knockdown in gynes. (**B**) The distance traveled by each individual for 10 min, 30 min, and 60 min. Compared to *ds-eGFP*-injected individuals, *ds-NPA*-injected individuals traveled shorter distances (7–9 individuals were assayed in 1 replicate, 3 replicates, which displayed different colors. Each dot represents an ant, *n* = 23 for *ds-eGFP* group and *n* = 23 for *ds-NPA* group, Wald test after fitting generalized linear model, *** *P* < 0.001). Bar represent mean and SEM. (**C**) The total foraging frequency and foraging duration after long-term dsRNA injection, indicating *NPA* knockdown induced lower foraging behavior than control group (6–9 individuals were assayed in 1 replicate, 3 replicates, which displayed different colors. Each dot represents an ant, *n* = 21 for *ds-eGFP* group and *n* = 22 for *ds-NPA* group, Wald test after fitting generalized linear model, *** *P* < 0.001). Bar represent mean and SEM. (**D**) The locomotion activity after long-term NPA peptide and BSA injection. Representative heatmaps shown locomotion trajectories of individuals in each group for 10 min, 30 min, and 60 min, indicating NPA peptide injection induced higher locomotion activities than control group. (**E**) The distance traveled by each individual for 10 min, 30 min, and 60 min, NPA peptide-injected queens traveled longer distances than BSA-injected queens (6–9 individuals were assayed in 1 replicate, 3 replicates, which displayed different colors. Each dot represents an ant, *n* = 20 for BSA-injected group and *n* = 22 for NPA peptide-injected group, Wald test after fitting generalized linear model, *** *P* < 0.001). Bar represent mean and

SEM. The numerical data for this figure can be found in S1 Data. BSA, bovine serum albumin; dsRNA, double-stranded RNA; NPA, neuroparsin A.

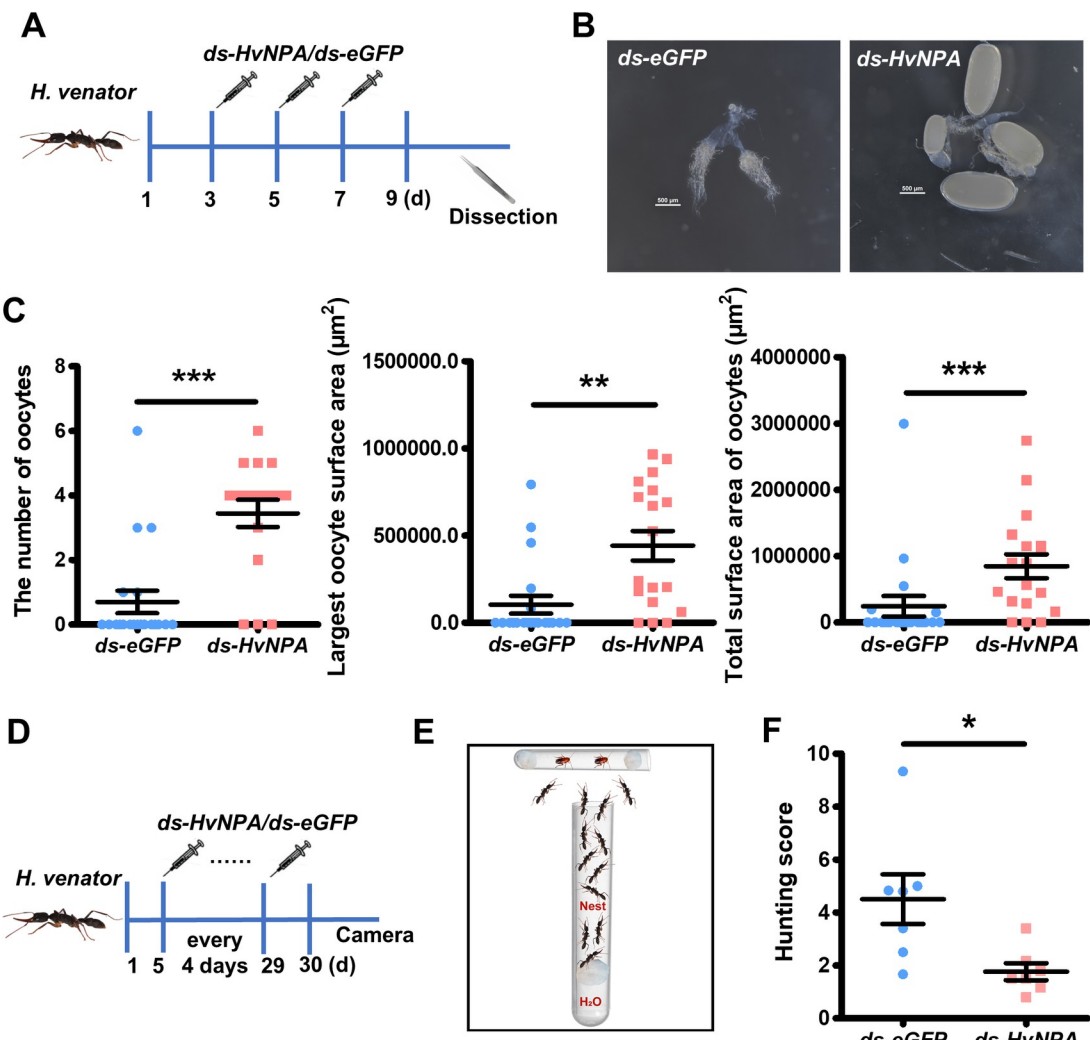

**Fig 4. Repression of NPA promotes gamergate-like behavior in *H. venator*.** (**A**) Schematic diagram of dsRNA injection for reproductive assay, the day when queen was removed from colony was labeled as the first day, dsRNA injection on day 3, 5, and 7, ovary dissection on day 9. (**B**) Representative ovaries of *ds-eGFP*- and *ds-HvNPA*-injected worker. Knockdown of *NPA* induced elevated ovary activity. (**C**) Scatter dot plot shown the oocyte number, the largest oocyte surface area, and the total surface area of oocytes of dsRNA-injected workers. *NPA* knockdown induced elevated ovary activities in workers. Bars represent mean and SEM. (*n* = 20 for *ds-eGFP* group and *n* = 18 for *ds-HvNPA* group, Mann–Whitney U test, two-tailed, ** *P* < 0.01, *** *P* < 0.001). Bar represent mean and SEM. (**D**) Schematic diagram of dsRNA injection for hunting assay. Workers were injected with dsRNA every 4 days after queen removal on day 1, 7 injections were performed in total for each worker, and hunting behaviors were recorded on day 30. (**E**) Schematic diagram of hunting assay. Half ants were injected with *ds-NPA* and other half were injected with *ds-eGFP*. The workers were introduced into a plastic box containing nest and transparent glass tube where 2 alive cockroaches were kept inside. The glass tube serves as the targets for hunting behavior. (**F**) The hunting score after dsRNA injection. Hunting behaviors include biting and tube wall touching. Hunting score was defined as the average of hunting behavior counts for each treatment. Each dot represents the hunting score of *ds-HvNPA*-injected or *ds-eGFP*-injected workers in 1 replicate, 7 replicates in total (Mann–Whitney U test, two-tailed, * *P* < 0.05). Bar represent mean and SEM. The numerical data for this figure can be found in S1 Data. dsRNA, double-stranded RNA; NPA, neuroparsin A.

assay, we labeled *ds-eGFP*-injected workers and *ds-HvNPA*-injected workers with different colors; 10 to 12 ants were introduced into the same rearing box, half from *ds-eGFP* group and half from *ds-HvNPA* group. Then, alive cockroaches were confined in a transparent test tube and introduced into the box (Fig 4E). We counted the hunting behavior as more than 4 s in touching the tube wall and an evident biting behavior. We videotaped the behaviors for 40 min. The hunting score was calculated as the average frequency of hunting behaviors of each individual of each group. We found that the *ds-HvNPA*-injected workers exhibited lower hunting score than the *ds-eGFP*-injected workers in the same box and showed a significant difference between the 2 groups across 7 different colonies assayed (Fig 4F). The above observations demonstrate that NPA signaling is causally involved in physiological and behavioral shift for *H. venator* worker/gamergate transition, suggesting the function of NPA in regulating gyne/queen transition is re-used in regulating worker/gamergate transition.

## NPA remotely affects ovary endocrine pathways

To investigate how NPA affects downstream molecular processes, we performed transcriptomics profiling for brain and ovary tissues when NPA level was either down-regulated or up-regulated in pharaoh ants. We injected *ds-eGFP/ds-NPA* or BSA/NPA (S3 Fig), then dissected brain and ovary for RNA-seq, and each brain or ovary was a biological replicate. For dsRNA injection, we analyzed 5 knockdown and 5 control brains, and 7 knockdown and 6 control ovaries. For peptide injection, we analyzed 7 NPA-injected and 10 control brains, and 10 NPA-injected and 12 control ovaries. In *ds-NPA*-treated brains, we found 41 DEGs, including 17 up-regulated genes and 24 down-regulated genes (adjusted $P < 0.05$) (S3 Data). Among the down-regulated genes, we detected *NPA*, indicating that the knock-down treatment was effective (S3 and S4 Figs). Besides *NPA*, we also identified the gene encoding pheromone-binding protein *Gp-9* (*general protein-9*, *LOC105835453*) as well as the gene encoding *KAZALD1* (*LOC105833561*), which is IGFBP-related gene, were significantly down-regulated (S3 Fig). In the NPA peptide-treated brain, we only detected 4 genes to be differentially expressed (adjusted $P < 0.05$) (S4 Data), suggesting that the brain was not sensitive to elevation of circulating NPA or the brain was slow in response (S3 Fig). In fire ants *Solenopsis invicta*, *Gp-9* is localized within a supergene which is crucial for determining the forms of the social organization of the colony [49]. *Gp-9* is probably responsible for carrying pheromones or transporting hormones in circulation [50]. In *H. saltator*, *Gp-9* in the brain was found to be remarkably down-regulated in reproductively activated gamergate; the observation is consistent with our finding that *Gp-9* was down-regulated in pharaoh ant queen brain [38]. KAZALD1 has an N-terminal IGFBP domain, besides that, it contains a serine peptidase inhibitor domain, a follistatin-like domain and an immunoglobulin-like domain, thus KAZALD1 may interact with insulin molecule and other proteins to function at localized brain regions for specific behavioral outputs [40,51].

In the *ds-NPA*-treated ovaries, we identified 77 DEGs, with 26 up-regulated DEGs, and 51 down-regulated DEGs (adjusted $P < 0.05$) (S5 Data). While in NPA peptide treated ovaries, we identified 1,051 DEGs, with 459 up-regulated DEGs, and 592 down-regulated DEGs (adjusted $P < 0.05$) (S6 Data). Twenty DEGs were shared between the 2 treatments, among which 10 DEGs were up-regulated in *dsRNA* treatment while down-regulated in NPA peptide treatment and other 10 DEGs displayed the reversed patterns (S4 Fig). Among these 20 shared DEGs, several genes were functionally related to nutrition and hormone pathways, including *yellow-g2* (*LOC105829555*), *shadow* (*LOC105831233*), *MRJP* (*major royal jelly protein*, *LOC105829551*), which exhibited negative correlation with NPA variation, as well as *JHBP* (*juvenile hormone binding protein*, *LOC105834712*) which exhibited positive correlation with

NPA variation (Figs 5A and S4). *Shadow* encodes a cytochrome P450 which is necessary for ecdysteroid synthesis while JHBP is potentially responsible for binding juvenile hormone and protecting it from degradation [52–54]. The results suggest that the brain secreted NPA remotely targets ovary endocrinological responses by affecting JH and ecdysteroid signaling in opposite directions. Yellow-g2 and MRJP belong to yellow/MRJP family which play crucial roles for reproductive maturation in honeybee [55]. Yellow-g2 was found in the Asian tiger mosquito to play a role for egg pigmentation and protect egg from desiccation [56]. It seems that these 2 genes are downstream effectors directly correlated with egg production machinery.

*JHBP* and *shadow* are shared DEGs from both dsRNA and peptide treatment with pronounced expression changes. To address whether *JHBP* is involved in reproductive regulation, we performed knockdown experiment by injecting *ds-JHBP* into hemolymph of gynes on day 4, 6, and 8 and found that *JHBP* down-regulation could significantly promote ovary activation, the number of yolky oocytes was increased and the sizes of yolky oocytes were notably enlarged (Fig 5B). These results indicate NPA suppression may act by down-regulating JHBP to induce reproductive activation. We further designed the HCR probes to detect the in situ expression of *JHBP* and *shadow* in the ovary and found their expression primarily distributed in ovary somatic cells (Figs 5C and S5). Our results indicate that the brain secreted peptide NPA could remotely regulate ovary somatic cell endocrine responses, oppositely affecting JH and ecdysteroid pathways to orchestrate the physiological and behavioral shifts (Fig 6).

## Discussion

Insemination triggers a repertoire of drastic physiological and behavioral changes so that the ant queen can realize its full reproductive functionality for the initiation of colony life. Gyne/queen transition implicates substantial shift of brain GRN in pharaoh ants and NPA is one of the earliest genes exhibiting differential expression in queens [6]. In this study, we validated the function of NPA in regulating reproductive plasticity for differentiated unmated queens and mated queens in pharaoh ants. Our results also suggest a potential role of NPA in regulating locomotion and foraging behaviors in ants. However, because we grouped multiple gynes/queens together in the same Petri dish for behavioral assays, we cannot exclude the effect of grouping variables that cause interdependency between data points. A more rigorous experimental design is needed to definitively determine the role of NPA in behavioral regulation. Furthermore, we found that NPA levels were consistently repressed in reproductively activated females across ant species and confirmed its function in worker/gamergate transition in ponerine ant *H. venator*. These results demonstrated the crucial mechanism of NPA in organizing a cascade of regulatory events for gyne/queen transition and the co-option of the mechanism in distantly related ant species for diversified scenarios of reproductive role differentiations when queens have lost the reproductive significance and workers can re-engage in reproduction.

### The anti-gonadotropic function of NPA in ants

NPA is found in diversified arthropods to be involved in reproductive regulation; however, its regulation is extremely variable and can be opposite in directions in different lineages [33,57–59]. In pharaoh ants, *NPA* is exclusively expressed in the MNSCs in the brain, co-localized with *ilp2*. *Ilp2* is found to be up-regulated in reproductive caste across ant species and ILP2 peptide treatment could promote ant reproduction [4,47]. NPA appears to play an antagonistic function to that of ILP2. NPA is homologous to vertebrate IGF binding protein (IGFBP) [26]. Most IGFs are bound with IGFBPs and transported to target tissues by circulation [60].

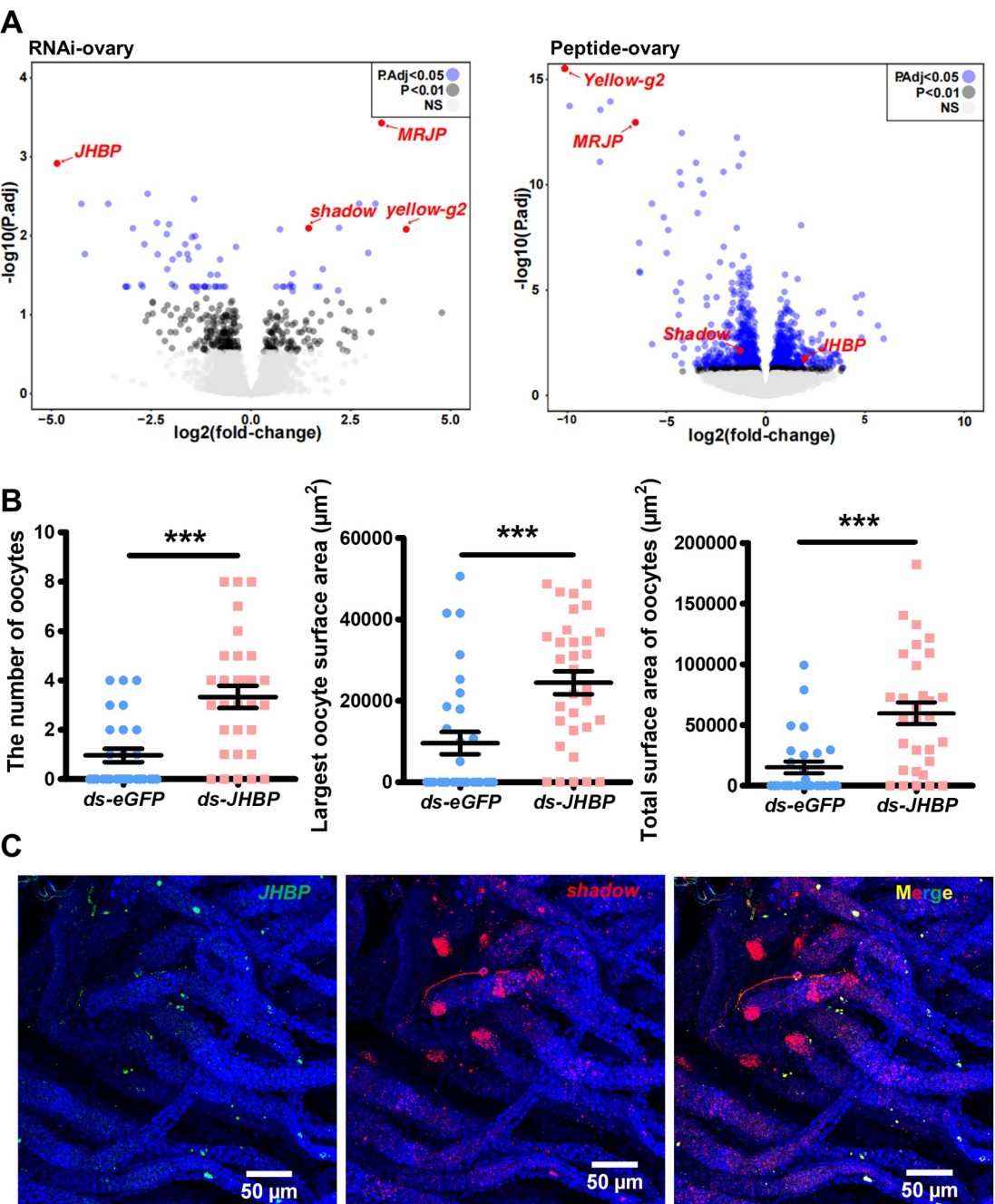

**Fig 5. Brain NPA remotely regulates ovary endocrine pathway related genes.** (A) Volcano plots of transcriptome from ovaries injected with dsRNA or peptide, genes with adjusted *p*-value <0.05 were colored in blue. Genes ranked top in DEG list and shown opposite responses to the 2 treatments were highlighted. RNAi-ovary data were from 6 and 7 biological replicates (individual ants) for *ds-eGFP* and *ds-NPA*, respectively. Peptide-ovary data were from 12 and 10 biological replicates for BAS and NPA peptide, respectively. (B) Scatter dot plots shown the oocyte number, the largest oocyte surface area, and the total surface area of oocytes of *JHBP* dsRNA-injected gynes and control gynes. *JHBP* knock down induced increased ovary activity (*n* = 28 for *ds-eGFP* group and *n* = 30 for *ds-JHBP* group, Mann–Whitney U test, two-tailed, *** *P* < 0.001). Bar represent mean and SEM. (C) In situ HCR staining of *JHBP* and *shadow* in the ovary. Scattered *JHBP* signals (green) were detected along ovarioles. *Shadow* signals (red) were concentrated in areas neighboring oviducts, scattered signals were also detected along ovarioles. The numerical data for this figure can be found in S1, S5, and S6 Data. DEG, differentially expressed gene; dsRNA, double-stranded RNA; HCR, hybridization chain reaction; NPA, neuroparsin A.

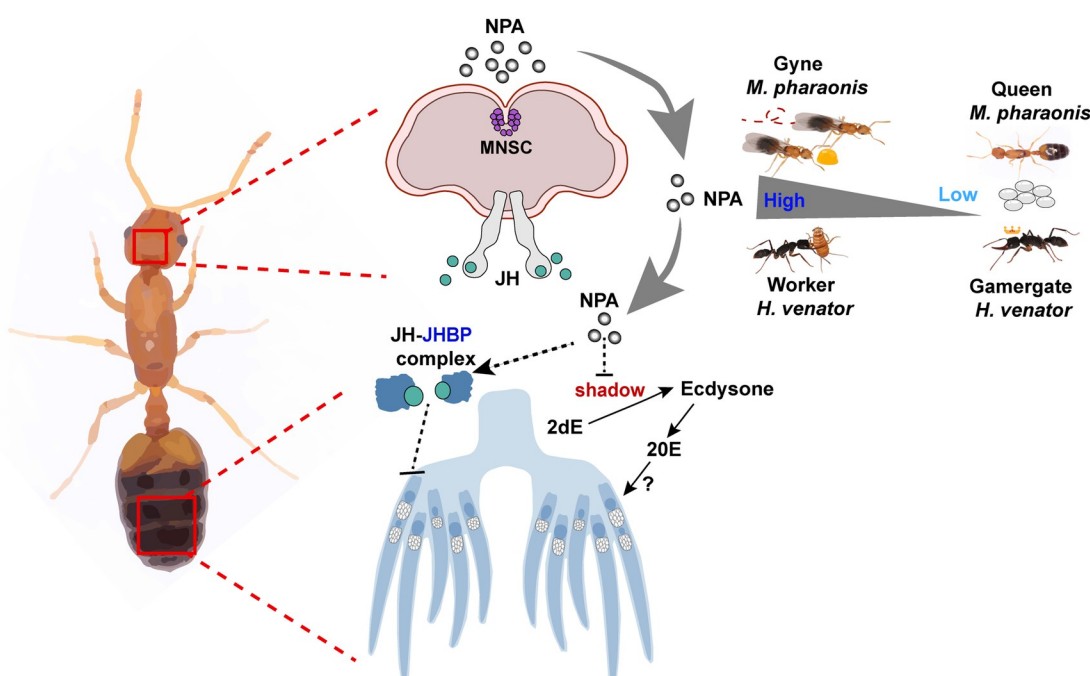

**Fig 6. Proposed model of the conserved function of NPA in regulating reproductive role differentiation across ant species.** Brain MNSCs secret NPA peptides into hemolymph of ants and remotely regulate ovary endocrine pathways. NPA positively regulates the expression of JHBP which binds and protects corpora allata secreted JH hormone from degradation while negatively regulates shadow which is required for ecdysone hormone synthesis. Higher NPA level induces higher JH while lower ecdysone signaling in the ovaries, thereby, suppresses ovary activity and promotes worker-like behaviors. Conversely, repression of NPA is necessary for the physiological and behavioral shifts into queen-like states. MNSC, medial neurosecretory cell; NPA, neuroparsin A.

IGFBPs could sequester the circulating IGF ligands globally or concentrate and cleave the ligands at localized sites [61]. IGFBPs also have IGF-independent functions, such as binding to surface receptors to trigger downstream signaling events or translocate into nucleus to directly regulate transcription [61]. There are multiple insulin molecules in ants and their functions in reproductive regulation are complex, depending on specific contexts or target tissues [38,47,62]. NPA is secreted from the brain into hemolymph, potentially buffering insulin molecules in circulation, thus antagonizing global insulin function in promoting reproduction in response to better nourishment. NPA may also act through other molecules, such as KAZALD1 to fine-tune insulin signaling in local tissues.

Apart from insulin signaling, NPA is also involved in endocrine pathways. In desert locusts, JH and ecdysteroid treatment could induce significantly altered expression of several *neuroparsin* transcripts [25]. Knock-down of JH receptor *methoprene-tolerant* (*met*) causes up-regulation of *SgNP3* and *SgNP4*, while knock-down of ecdysone receptor complex causes downregulation of *SgNP3* and *SgNP4* [29–31]. In migratory locust, injection of anti-serum against NPA could recapitulate the effect of JH injection in pigmentation, metamorphosis, and vitellogenesis, suggesting that NPA is an inhibitory factor for corpora allatum JH system, antagonizing JH effect both for development and reproduction [33]. As for ecdysone signaling, NPA seems to play a facilitative role. As indicated by explant culture of locust embryonic central nervous system, ecdysone is found to be a necessary factor for the outgrowth of neurites and NPA application could promote ecdysone effect synergistically [63]. In pharaoh ants, we found that NPA plays an anti-gonadotropic function, and the function may largely be attributed to altered endocrine signaling. Our findings demonstrate that NPA potentially acts in concert with JH signaling by increasing JHBP level while acts in opposite with ecdysone

signaling by decreasing ecdysone synthesis. Together, these results demonstrate the complex interactions and potential feedback relationships between insulin, NPA, and endocrine signaling pathways. The sophisticated re-configuration of the crosstalk of these core components may shape the reproductive behaviors of different insects for adapting to diversified environments and life history choices.

## NPA initiates a cascade of molecular events underlying reproductive role differentiation

Besides NPA, there are several other brain neuropeptide genes that are also differentially expressed when a pharaoh ant gyne transitions to a queen, including the down-regulation of corazonin, tachykinin and up-regulation of drosulfakinin [6]. It is of interest that all these neuropeptides potentially have cross-talks with insulin signaling pathways. In fruit flies, the receptors of corazonin and tachykinin are detected in insulin-producing cells, causally affecting insulin signaling pathways [39,64,65]. Drosulfakinin is found to negatively feedback to insulin expression [66,67]. Consistent with the anti-correlation relationship between corazonin expression and reproductive activation in pharaoh ants, corazonin plays a conserved role for worker/gamergate transition in *H. saltator*, suppressing vitellogenesis while promoting worker behaviors [68]. Of these neuropeptide genes, NPA is the earliest to exhibit suppression in queen compared to age-matched gyne, then followed by altered expression of other peptide genes [6]. The observation suggests that suppression of NPA is potentially responsible for initiating a cascade of signaling events and gradually implicating more genes, as a result of which, a directional physiological and behavioral shift is stabilized for differentiated reproductive roles.

By profiling brain transcriptome in response to NPA changes, we found that NPA positively regulates a pheromone-binding protein which is homologous to *Solenopsis invicta Gp-9*. *Solenopsis invicta Gp-9* is localized within a recombination-suppressed supergene-like region together with hundreds of other genes and the allelic variation of this region is critically involved in social organization, determining whether workers accept 1 queen or multiple queens in a colony [49,50]. In *H. saltator*, when a worker transitions to a gamergate, the olfactory response to pheromones is reduced, potentially releasing the gamergate from mutual inhibitory or self-inhibitory effect of pheromones on reproduction [38]. Similarly, in *Camponotus floridanus* ants, the differences of behavioral spectrum between minor and major workers are profoundly correlated with their olfactory sensitivities, with majors displaying limited behavioral repertoire and generally reduced olfactory sensitivity [69]. In the same sense, pharaoh ant queens display extremely stereotyped behaviors which might partially be due to the impaired pheromonal sensation caused by *Gp-9* down-regulation.

We also examined the ovary expressed genes that are sensitive to NPA changes, among which the gene *JHBP* and *shadow* consistently respond when NPA is either increased or decreased. *NPA positively regulates JHBP while negatively regulates shadow*. JHBP is potentially responsible for binding juvenile hormone and protecting it from degradation while *shadow* is necessary for ecdysteroid synthesis. Thus, it seems that NPA suppression caused a sustained decreased JH and increased ecdysteroid signaling state. We further showed that disruption of *JHBP* had similar effect as disruption of *NPA* in promoting vitellogenesis, consistent with anti-gonadotropic function of JH as reported in pharaoh ants previously [70,71]. The high ecdysteroid/low JH signaling states are also correlated with reproductive activation in *H. saltator* gamergates [13,38,72]. High ecdysteroid is found to drive gamergate fate while high JH is found to drive worker fate [13,38]. Ecdysteroid and JH act through transcription factor *Kr-h1* to bind and repress different set of genes, stabilizing distinct brain functions for corresponding reproductive roles [13]. Thus, multiple observations converge to suggest that the neuro-hormonal

signaling cascade for reproductive role differentiations is shared by diversified ant species and NPA serves a conserved and crucial function in initiating the signaling of the cascade.

## The PI-CA-ovary axis in orchestrating physiological/behavioral shifts in ants

Analogous to HPG axis in orchestrating reproductive physiological and behavioral shifts in mammals, in insects, female reproduction is primarily regulated by the pars intercerebralis (PI)–corpora allatum (CA)–ovary axis [73]. Pars intercerebralis secret an array of neuropeptides, among which insulin is involved in nutrition response, usually responsible for promoting reproduction while limiting lifespan [74]. Corpora allatum secret JH which normally plays gonadotropic functions, promoting vitellogenesis process in many insects. The function of JH in reproduction is less significant in some insects while ovary secreted ecdysteroid plays a major role in reproduction [75].

Here, we demonstrate that the PI–CA–ovary axis is responsible for orchestrating a repertoire of physiological and behavioral shifts underlying reproductive role differentiation in ants. The activity of the axis can be modulated by multiple external stimuli for different reproductive role differentiation scenarios. In *M. pharaonis* gyne/queen transition model, the external stimulus is insemination event; in *H. saltator* worker/gamergate transition model, the external stimulus is absence of queen pheromones; and in *O. biroi* un-reproductive/reproductive transition model, the external stimulus is absence of larval signals [6,8,47]. Pars intercerebralis receive inputs of diversified neuromodulators and peptides, in fruit flies it integrates signals from multiple tissues such as intestine, fat body, and corpora cadiaca [74]. Thus, different environmental stimuli may firstly activate diversified sensory neurons, which then transduce the signals onto pars intercerebralis. The pars intercerebralis is the center integrating multiple inputs, then responds by releasing neuro-hormones accordingly, finally triggers a cascade of molecular events for physiological and behavioral remodeling.

The status quo functions of insulin and JH in regulating reproduction appear to be re-wired in social insects. In honeybees, higher nutrition is correlated with lower insulin level and queens have reduced insulin level [76]. The conventional gonadotropic function of JH appears to be reversed in social insects, in which lower titer of JH is correlated with higher reproduction and treatment of JH analog inhibits vitellogenesis, causing reduced egg production in female reproductive castes [62,70,71,77–79]. In *H. saltator* ants, it is shown that insulin signaling is diverged into 2 branches, activating either the mitogen-activated protein kinase (MAPK) or Akt/FOXO pathway which is involved in regulating reproduction or lifespan, respectively, thus decoupling the trade-off relationship of fecundity and longevity observed in a wide array of insects [4]. In mosquitoes, NPA homolog binds to a receptor tyrosine kinase and act in concert with ILP3 to activate Akt signaling pathways in follicle cells for ecdysteroid synthesis [57,80]. Opposite to that in mosquitoes, we have demonstrated a suppressive function of NPA in ecdysteroid synthesis in ants, suggesting that NPA function is also re-wired in ants. The sophistication of how NPA, insulin, and endocrine pathways interact to function in a context-dependent and tissue-specific manner for reproductive role differentiations and the significance of the re-wirings of their functions for life history choices in ants remain to be further interrogated.

## Materials and methods

### Biological samples

*M. pharaonis* was originally collected in June 2016 from Xishuangbanna, Yunnan province, China. New colonies were established by split from the original one. Ants were reared at 27˚C,

65% RH, and a 12h:12h light:dark cycle. Pharaoh ants were fed with shredded *Tenebrio molitor* and customized diet every 2 days. The customized diet was made of 60 g sucrose (Sangon, A502792-0500), 6 g agar (Sangon, A505255-0250), and 2 tablets of vitamins with minerals dispersible tablets 21 (Kangkang, 10051475686242) heated and dissolved in 500 ml ddH$_2$0. The liquid diet is then poured into Petri dishes to cool down, then stored at 4˚C.

*H. venator* was collected in July 2022 from Quanzhou, Fujian province, China. All colonies were housed in plastic boxes and fed with cherry-red cockroaches (*Blatta lateralis*) every 3 days.

*O. biroi* was collected from Zhongshan, Guangdong province, China. Ants were housed in sterilized soil and fed with *T bicarinatum* larvae and pupae every 3 days.

*T. bicarinatum* gynes and queens for experiment were collected from Quanan, Fujian province, China. The alive ants were flash frozen by liquid nitrogen and stored at −80˚C refrigerator for RNA extraction.

*C. japonicus* were collected from Weifang, Shandong province, China.

*C. nicobarensis* were collected from Sanming, Fujian province, China.

*G. bicolor*, collection place was not recorded, Hao Ran, a myrmecological researcher, identified the species as *G. bicolor* based on its morphological features.

*O. smaragdina* were collected from Huazhou, Guangdong province, China.

*L. alienus* were collected from Zhangjiakou, Hebei province, China.

## RNA interference

The specific dsRNA for interfering *NPA* (*LOC105837629*), *JHBP* (*LOC105834712*), and *HvNPA* were designed on the E-RNAi website (www.dkfz.de/signaling/e-rnai3/), *eGFP* and *NPA*-random dsRNA was used as control. The randomized sequence of *NPA* was designed on http://www.detaibio.com/sms2/random_dna.html, which used the original *NPA* CDS as template to obtain a shuffled sequence of *NPA*, and blast the generated sequence was non-target to *M. pharaonis*. Chemically synthesized *NPA-random* sequence was ligated into a vector, which was then used as a template to generate ds-NPA-random in vitro. dsRNA was synthesized using MEGAscript RNAi kit (Thermo Fisher AM1626) for the target genes. After digestion of the DNA template and single stranded RNA, and washing away the residuals, synthesized dsRNA was eluted by RNase-free water to a concentration of 7 μg/μl. Purified dsRNA was microinjected into the hemolymph by using micro-injector (Eppendorf Femtojet). The injected volume of dsRNA was approximately 0.1 μl. The primers used to amplify dsRNA templates were as following Table 1:

## dsRNA and peptide injection

dsRNA or peptide were microinjected into hemolymph using micro-injector (Eppendorf Femtojet) with glass capillary needle (Beijingzhengtian BJ-40) through the cuticular hole which

**Table 1. The primers for amplification of dsRNA templates.**

| Gene | Forward primer | Reverse primer |
| --- | --- | --- |
| *NPA* | taatacgactcactatagggTATCTTCCAACTCACTCACGC | taatacgactcactatagggTCACATTCGCTCCTCGC |
| *NPA-random* | taatacgactcactatagggGGAAAAAGCGGGAAAGTTCT | taatacgactcactatagggTGCCCCAAAGAGTGAATAGC |
| *JHBP* | taatacgactcactatagggGACTTTCCACCTTTGGAGCC | taatacgactcactatagggATATATCAGCCGTTGCTGGC |
| *HvNPA* | taatacgactcactatagggACTGTGGAGGTCCGAGTCAG | taatacgactcactatagggTCTCAATCGTCGTTCATCCA |
| *eGFP* | taatacgactcactatagggAGTGCTTCAGCCGCTACCC | taatacgactcactatagggCATGCCGAGAGTGATCCCG |

was pierced by a tungsten needle. We choose the dorsal side of thorax as injection site in *M. pharaonis* and dorsal side of the head for *H. venator*.

For reproduction assay of *M. pharaonis* ant, we collected gyne pupae from multiple subcolonies, pooled the gynes that eclosed on the same day in a new plastic box, where food and water were provided. The 4-day-old gynes were then divided randomly into experimental and control groups, the experimental group were injected with dsRNA for target genes and the control groups were injected with *ds-eGFP* and *ds-random*. *H. venator* workers were injected with *ds-NPA* as treatment and injected *ds-eGFP* as control. The ants were injected every 2 days, 3 times in total for *M. pharaonis* and *H. venator*. In peptide injection manipulation, for each experimental and control group, we collected approximately 60 newly eclosed gynes and approximately 90 males on day 0; we let them mate on day 3, remove the males on day 4, then inject NPA peptide/BSA/heat-inactivated NPA on days 5 and 7. We excluded the individuals that are not successfully inseminated from our analysis (by checking the spermatheca). The concentration of the injected NPA peptide is 600 ng/ml, and the same concentration NPA peptide was incubated at 100°C for 10 min to be denatured. Heat-inactivated NPA as well as BSA were used as control groups in this experiment. The peptides were injected into the hemolymph from the dorsal side the thorax every 2 days for 2 times.

For reproduction assay of *H. venator* ants, workers from the same colony were injected with *ds-NPA* as treatment and injected *ds-eGFP* as control. The ants were injected every 2 days, 3 times in total.

For *M. pharaonis* locomotion and foraging behaviors, we collected gyne pupae from multiple subcolonies, pooled the gynes that eclosed on the same day (20 to 30 individuals for 1 experimental group) in a new plastic box, and randomly separated the gynes into 2 groups. For dsRNA treatment, one group were injected with *ds-NPA*, the other group were injected with *ds-eGFP*. Four-day-old gynes were injected with dsRNA every 4 days, 7 times in total; 30-day-old gynes were used for behavioral assay (S2 Fig). For peptide treatment, the gynes were allowed to mate on day 3, then one group were injected with NPA peptide, the other group were injected with BSA. Six-day-old queens were injected with peptide every 4 days, 6 times in total; 30-day-old queens were used for behavioral assay (S2 Fig).

For *H. venator* hunting assay, workers from the same colony (queen absent) were divided into 2 groups and labeled with different colors. One group was injected with *ds-HvNPA* and the other group was injected with *ds-eGFP*, injection was performed every 4 days, lasting for 30 days. Seven biological replicates were used.

## Ovary dissection and measurement

Ants were anaesthetized on ice and were dissected in ice-cold phosphate-buffered saline (PBS, Solarbio, P1000; DEPC, Sangon, B600154-0100; pH 7.5) under a stereomicroscope (Nikon SMZ800). We dissected the ovary for measurement using 9-day-old gynes or queens in *M. pharaonis*. In *H. venator*, dissection was performed 9 days after the first day of injection. The ovaries were dissected in DEPC-treated PBS (pH 7.5), then imaged by Nikon microscope (Nikon SMZ18). The number of yolky oocytes was counted and the surface areas of the yolky oocytes were measured using manufacturer provided software (NIS-Elements D 5.10.00 64-bit).

## RNA extraction and qPCR

Ant RNA was extracted with Trizol (Thermo Fisher 15596018) from a single brain, ovary, or head (s). For qPCR, the cDNA was reverse transcribed by PrimeScript RT reagent Kit with gDNA Eraser (Takara RR047A); 100 to 1,000 ng total RNA was used. qPCR was performed on

**Table 2.  The primer list for quantifying *NPA* and reference genes across ant species.**

| Gene | Forward primer | Reverse primer |
|---|---|---|
| *NPA* | AGATAGCTCATCGGAGCCC | CACAGAAATGCCATTATACACG |
| *MpEF1A* | TTCATTTATTGCTCTCACATCTACG | ACCGTTGCCCTTTCTACTCTAA |
| *HvNPA* | ACGATCAAAAGAGTGCACGG | CATCCCATGACTGACTCGGA |
| *Hv-RPL32* | CCAACTGGCTTCCGAAAAGT | CACACGTATTGAGAGCTGTCG |
| *ObNP* | ATTCAGCCAAGCGATCGTTC | TTTATCACATTCGTCGCCGC |
| *Ob- RPL32* | TCCTCATGATGCAGAACCGT | TTCAACGATGGCCTTCCTCT |
| *TbNPA* | GAAAAGGTCCCGGGCAAATT | GCAGACAGGTATTCGCGAAG |
| *Tb-RPL32* | CGATTCGACCAGTCTACCGA | GCCAGTTGCGCTTGAGTTTA |
| *GbNPA* | GCTTTTCTGGCTGCAATCCT | ATCACATTCATCGCCACAGC |
| *Gb-RPL32* | TAAACTCAAGCGGAACTGGC | CCAGTTGGTAGCATGTGACG |
| *CnNPA* | CCTTTCTGGCCGCAATCTTT | TATCACATTCGTTGCCGCAG |
| *Cn-RPL32* | GAAAACACGTCACATGCTGC | CACTGGCATTTGTCACACGT |
| *CjNPA* | TCTGGCCGCAATCTGTCTAA | TTTATCACATTCGTCGCCGC |
| *Cj-RPL32* | CGTCACATGCTACCAACAGG | CTGGCATTTGTCACGCGTAT |
| *OsNPA* | GGCGATGGATTGATCTGCAG | AGAGTCTGATGGGGCAAACA |
| *Os-RPL32* | CGTCACATGCTACCAACAGG | AAGTTGTTGAGCACGTTCGA |
| *LaNPA* | GTGTGTGGCGACGGATTAAT | TGATGGGGCAGACAAGGATT |
| *La-RPL32* | CGTTTCAAGGGCCAGTACTT | GCTCGACGATGGCTTTTCTT |

Roche LightCycler96 with TB Green Premix Ex Taq II (Takara RR820A). *M. pharaonis* house-keeping gene *EF1A* (*LOC105833278*) was used as endogenous reference, *RPL32s* were used as endogenous reference genes for other ant species to normalize to the expression level of target genes [81]. The primers used to do qPCR were as following Table 2:

## RNA-sequencing library construction and data analysis

For brain/ovary transcriptomics analysis, we collected the gynes that are eclosed on the same day from multiple colonies, pooled them together, and then randomly separated the gynes into 2 groups. For transcriptomics analysis after *NPA* knockdown, we injected one group of gynes with *ds-NPA* on day 4, day 6, and day 8, and dissected their brains/ovaries on day 9 for transcriptomics analysis. While we injected the other group of gynes with *ds-eGFP* following the same protocol with a comparable amount of dsRNA and dissected their brains/ovaries on day 9 for transcriptomics analysis. For transcriptomics analysis after peptide treatment, we allowed the gynes to mate with males on day 3 after eclosion. On day 4, we removed the males, then injected one group of queen with NPA peptide on day 5 and day 7, and dissected their brains/ovaries on day 9 for transcriptomics analysis. While we injected the other group of queens with BSA following the same protocol with a comparable amount of proteins and dissected their brains/ovaries on day 9 for transcriptomics analysis (S3 Fig).

Library construction and sequencing were performed by BGI, Shenzhen, China. *M. pharaonis* gyne/queen RNA samples were extracted from a single brain or ovary with Trizol. For library construction, 5 ng RNA was used to be reverse transcribed to cDNA, then PCR was performed for pre-amplification. PCR products were fragmented and Tn5 adaptor was added for amplification. The amplified products were purified, then circularization and exonuclease digestion were performed.

RNA sequencing was performed by BGI's DIPSEQT1; SOAPnuke 1.5.6 was used for quality control to filter out low-quality sequences and adapter sequences [82]. RNA-seq reads were mapped to the reference *M. pharaonis* genome using STAR 2.7.6a with default parameters

[83,84]. Differential expression was analyzed with the DESeq2 package, we used the Wald test to get the *P*-value for each gene, and the *P*-values are corrected for multiple testing using the default Benjamini and Hochberg method [85]. For the GO enrichments of peptide-injected ovarian DEGs, we used clusterProfiler 4.2.2 with multiple test correction to correct the *P*-value and choose Benjamini and Hochberg correction method. GO terms for each gene were annotated according to their homologous relationship to *Drosophila melanogaster*, *Caenorhabditis elegans*, *Homo sapiens*, and Uniprot Database. The raw RNA-seq data of brains and ovaries generated in this study are deposited in the CNGB Nucleotide Sequence Archive (CNSA) with accession number CNP0004568.

## Behavioral assays

The *M. pharaonis* behavioral experiments were performed in a temperature and humidity-controlled room. dsRNA or peptide injection treatments were detailed in the previous paragraph entitled "dsRNA and peptide injection", and 7 to 9 gynes or queens were transferred to a 9 × 9 cm Petri dish (Biosharp, BS-90-D) for behavioral assays. The Petri dishes of the experimental and control groups were positioned neighboring to each other (<20 cm) during behavioral recording, and their behaviors were videotaped during the same period of time. Their activities were recorded by video recorder (Chishan, IK-H5H5-9A8L2). The first 60-min videos were used to track individual locomotion trajectory. Then, a patch of food was placed on aluminum foil and was introduced into the Petri dish; for the following 40 min, foraging behavior was recorded.

We used the following protocol to analyze locomotion behavior: (i) installing Anaconda on 64-Bit Windows, open Anaconda and using installer program, pip, to install opencv-python, numpy, pandas, matplotlib, seaborn, and ast, then install trajectory analysis package, trackerao; (ii) using Corel Video studio 2021 to accelerate a video to quadruple speed; (iii) trajectory tracking was conducted in Adobe After Effects CC 2019 built-in tracker, choose track motion for individual tracking, then edit track target as a new object and link a layer. Each individual and pseudo-nest were represented by color square (Lime, R, G, B = 0, 255, 0; Navy, R, G, B = 0, 0, 128; Magenta, R, G, B = 255, 0, 255), respectively. Since 7 to 9 ants were contained in 1 Petri dish, color square setting was performed 3 to 4 times for different ants in the video. Nest as an anchor point for ants in the Petri dish, and 2 to 3 individuals were tracked simultaneously. The tracking video was exported, displaying 3 colored squares with white background by Adobe Media Encoder CC 2019. All exported files were transformed from MP4 to AVI format in Format Factory; (iv) open Anaconda Powershell Prompt entering video path and then typing jupyter notebook for remote connection. Establish a new python3 notebook on the web page, entering codes for starting, then enter processed video path and name (S1 Text). Input squares number and pitch up color squares in sequence in the new ROI frame. The information of pseudo-nest and ant motion coordinates were saved in the same file with processed video after the squares tracking finished; (v) establish a new python3 notebook for figure generation and coordinates statistics. Entering parameter setting code to set up appropriate parameters for dots per inch (dip), figsize, Vmax, min-distance, and distance, then operate codes (S1 Text). At last, entering figure generation codes and waiting heatmap created (S1 Text).

For foraging analysis in *M. pharaonis*, we defined an effective forage as an individual contacted food for more than 3 s. We counted the frequency and duration of foraging behaviors.

In *H. venator*, hunting behavior was assayed. We painted individuals with 2 different colors, *ds-eGFP*-injected ants were painted blue and *ds-HvNP*-injected ants were painted white, and 10 to 12 ants were introduced into the same plastic box, 5 to 6 from *ds-eGFP* group and 5 to 6 from *ds-HvNPA* group. Alive cockroaches were confined in a transparent test tube. An

effective hunting behavior was defined in reference to *Harpegnathos saltator* hunting behavior, as an individual spent more than 4 s in touching the tube wall and an evident biting behavior [68]. The hunting score was calculated as the mean frequency of hunting behaviors of the 5 to 6 individuals from the same treatment group.

## Hybridization chain reaction (HCR)

The HCR probes used for detecting *NPA* in the brains of *M. pharaonis*, *H. venator*, and *O.biroi* and for detecting *JHBP* and *shadow* in the ovary of *M. pharaonis* were designed according to instructions in the website https://github.com/rwnull/insitu_probe_generator.

The brains and ovaries were dissected in DEPC-PBS and fixed in ISH Fixative Solution (Servicebio, G1113-500ML) for 20 min, and then we generally followed the protocol provided by Molecular Instrument (www.molecularinstruments.com). Alexa Flour 488 was used for detection of *NPA* in *M. pharaonis*, Alexa Flour 546 were used for *NPA* detection in *H. venator* and *O.biroi*. Alexa 488 and 546 were used for *JHBP* and shadow detection in ovary, respectively. The brains and ovaries were imaged with a confocal microscope (Nikon, A1 MP+). Images modify were used ImageJ for channel merge, split, and set scales.

## NPA peptide synthesis

The protein sequence of NPA was downloaded from NCBI. The sequence of 38-135aa was selected for synthesis. The selected sequence was truncated of signal peptide and contains 12 conserved Cystein residuals. The corresponding NPA peptide coding sequence was PCR amplified from cDNA and was inserted into pGEX-4T-AB1 expression vector. Transformation was performed using *Escherichia coli* Rosetta. Recombinant NPA was induced under 0.8 mM IPTG (Sangon, 367-93-1) and 100 μg/ml ampicillin (Sangon, 7177-48-2) in Lysogeny broth liquid medium (Sangon, A507002-0250) at 37°C for 4 h. SDS-PAGE (Bio-Rad, 1615100) analysis indicated that the recombinant NPA was expressed in the supernatant. Recombinant NPA was harvested and purified in 250 mM imidazole solution (AMRESCO/VWR, 0527-100G). Recombinant NPA was dissolved in PBS (pH 7.3) to a concentration of 1 mg/ml with 85% purity, 10% glycerol was added to the solution. The peptide was kept at −80°C for long-term storage. The peptide was diluted with PBS buffer to a working concentration of 600 ng/ml.

## Statistical analysis

All scatter plots were generated by GraphPad Prism 5. For qPCR analysis, we compared the differences between 2 groups using Mann–Whitney U test (two-tailed). We compared transcripts per million (TPM) of *NPA* between *ds-NPA-* and *ds-eGFP*-injected brain samples using Mann–Whitney U test (two-tailed). For reproductive data of *M. pharaonis* and *H. venator*, we compared the differences between 2 groups using Mann–Whitney U test (two-tailed). For *H. venator* hunting scores, we compared the differences between 2 groups using Mann–Whitney U test (two-tailed). For *M. pharaonis* behavioral data, the behavioral statistical analyses were performed in an R (v4.1.2) environment. We considered that the foraging and locomotion behaviors in a given time following compound Poisson processes, where behaviors came with associated variables such as duration or walking distance. We employed the R package "cplm" of which the zero inflation of our data can be handled to fit a null model (mod0 = cpglm (Value~Replicate)) and an alternative model (mod1 = cpglm(Value~Replicate+Treatment)) [86]. The alternative and the null model were then compared using the waldtest function of the R package "lmtest" and the *P*-value was calculated [87].

## Supporting information

**S1 Fig. Co-expression of NPA and Ilp2 in the brain of *M. pharaonis* and NPA multiple alignments.** (A) HCR staining and immunofluorescence shown *NPA* mRNA and peptide localization in the medial neurosecretory cells (MNSC). Immunofluorescence of Ilp2 shown its co-expression with NPA. Magenta represents *NPA* mRNA, red NPA peptide, green Ilp2 peptide, and blue DAPI. (B) Sequences aligned by clustal W, performed in MEGA11, and decorated by ESPript 3. Conserved cysteines were indicated in white bold font and highlighted in red. Signal peptides were highlighted in yellow. Ant, bee, wasp, and termite NPA contain 14 cystine residuals, locust NPA and mosquito OEH had 12 cystines, human IGFBP had 16 conserved cystines. These sequences were downloaded from NCBI with the following accession numbers. *M. pharaonis*: XP_012538010; *O. biroi*: XP_011333498; *H. saltator*: XP_025154127; *A. mellifera*: XP_026296327.1; *V. mandarinia*: XP_035744017.1; *S. gregaria*: CAC38869.1; *L. migratoria*: CAA76829.1; *Z. nevadensis*: KDR17790.1; *A. gambiae*: XP_311039.2; *H. sapiens*: KAI4013757.1.
(TIF)

**S2 Fig. Behavior analysis of *M. pharaonis*.** (A) The schematic diagram for behavioral assay after dsRNA injection. (B) Foraging analysis in blank area after multiple dsRNA injection as control (6–9 individuals were assayed in 1 replicate, 3 replicates, which were labeled by different colors. Each dot represents an ant, *n* = 21 for *ds-eGFP* group and *n* = 22 for *ds-NPA* group, Wald test after fitting generalized linear model, no significance). (C) The schematic diagram for behavioral assay after peptide injection. (D) Scatter dot plots shown foraging frequency and foraging duration for NPA-injected and BSA-injected groups (6–9 individuals were assayed in 1 replicate, 3 replicates, which were labeled by different colors. Each dot represents an ant, *n* = 20 for BSA-injected group and *n* = 22 for NPA peptide-injected group). (E) Foraging frequency and duration analysis in blank area after multiple peptide injection as control (6–9 individuals were assayed in 1 replicate, 3 replicates, which were labeled by different colors. Each dot represents an ant, *n* = 20 for BSA-injected group and *n* = 22 for NPA peptide-injected group). The numerical data for this figure can be found in S2 Data.
(TIF)

**S3 Fig. Transcriptomics analysis upon *NPA* knockdown and peptide injection.** (A) The tissue-specific expression of *NPA* in *M. pharaonis* quantified by qPCR. (B) RNAi efficiency of *NPA* 24h after dsRNA injection, the manipulation repeated 2 times. (C) Schematic diagram for RNA-seq sample collection. (D) Volcano plot of transcriptome of gyne brains. The gynes were injected with dsRNA every 2 days, 3 times. Genes with adjusted *P* < 0.05 are highlighted in blue. Data are from 5 biological replicates (individual ants) per treatment. (E) Volcano plot of transcriptome of queen brains. The queens were mated on day 3 and injected NPA or BSA on days 5 and 7. Genes with adjusted *P* < 0.05 are highlighted in blue. Data analysis from 7 NPA-injected and 10 control brains. The numerical data for this figure can be found in S2 Data. Transcriptome data for this figure can be found in S3 and S4 Data.
(TIF)

**S4 Fig. Differentially expressed genes identified by transcriptomics analysis in *M. pharaonis*.** (A) *NPA* transcripts per million (TPMs) in the brains after multiple dsRNA injections (*ds-NPA* sequence were removed from transcriptomics analysis). (B) Analysis of ovary DEGs with dsRNA and peptide injections; 20 DEGs are shared between dsRNA-injected and peptide-injected groups. Among the shared 20 DEGs, 10 DEGs are up-regulated in ds-NPA-injected group while down-regulated in NPA peptide-injected group, including *shadow*, *MRJP*, and *yellow-g2*. Another 10 DEGs are down-regulated in *ds-NPA*-injected group while up-regulated in

NPA peptide-injected group, including *JHBP*. (C) Predicted domain structures of JHBP, KAZALD1, yellow-g2, and MRJP, SP indicates signal peptide. (D) GO enrichment analysis of down-regulated genes in peptide-injected ovaries. (E) GO enrichment analysis of up-regulated genes in peptide-injected ovaries. The numerical and GO data for this figure can be found in S2 and S7 Data, respectively.
(TIF)

**S5 Fig. HCR staining of *vasa* and *shadow* in the ovary.** *Shadow* and *vasa* were primarily expressed in distinct segments in the ovarioles. Green, red, and blue represent *vasa*, *shadow*, and DAPI, respectively.
(TIF)

**S1 Movie. *M. pharaonis* foraging behavior.**
(MP4)

**S2 Movie. *H. venator* wall touching behavior.**
(MP4)

**S3 Movie. *H. venator* biting behavior.**
(MP4)

**S1 Data. The numerical data of Figs 1–6.**
(XLSX)

**S2 Data. The numerical data of S1–S5 Figs.**
(XLSX)

**S3 Data. All DEGs *ds-NPA* injection vs. *ds-eGFP* injection in brain.**
(XLSX)

**S4 Data. All DEGs NPA peptide injection vs. BSA injection in brain.**
(XLSX)

**S5 Data. All DEGs *ds-NPA* injection vs. *ds-eGFP* injection in ovary.**
(XLSX)

**S6 Data. All DEGs NPA peptide injection vs. BSA injection in ovary.**
(XLSX)

**S7 Data. GO enrichments of ovary DEG in peptide injection treatment.**
(XLSX)

**S1 Text. Locomotion analysis codes.**
(TXT)

**S1 Zip. dsRNA-injected locomotion trajectory coordinates.**
(RAR)

**S2 Zip. Peptide-injected locomotion trajectory coordinates.**
(RAR)

**S3 Zip. Scripts.**
(RAR)

## Acknowledgments

We would like to thank Cong Li for his technical supports of Confocal Microscopy.

## Author Contributions

**Conceptualization:** Xiafang Zhang, Weiwei Liu, Guojie Zhang.

**Data curation:** Xiafang Zhang, Nianxia Xie, Pei Zhang, Qiye Li.

**Formal analysis:** Xiafang Zhang.

**Funding acquisition:** Weiwei Liu, Guojie Zhang.

**Investigation:** Xiafang Zhang, Nianxia Xie, Wei Dai, Wenjiang Zhong, Dashuang Zuo, Jie Zhao, Pei Zhang, Weiwei Liu.

**Methodology:** Xiafang Zhang, Nianxia Xie, Guo Ding, Dongdong Ning, Wei Dai, Pei Zhang, Chengyuan Liu, Qiye Li.

**Project administration:** Qiye Li, Weiwei Liu, Guojie Zhang.

**Resources:** Chengyuan Liu, Hao Ran.

**Software:** Nianxia Xie, Dongdong Ning, Chengyuan Liu, Qiye Li.

**Supervision:** Weiwei Liu, Guojie Zhang.

**Validation:** Xiafang Zhang.

**Visualization:** Xiafang Zhang, Zijun Xiong.

**Writing – original draft:** Xiafang Zhang.

**Writing – review & editing:** Xiafang Zhang, Weiwei Liu, Guojie Zhang.

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
