## [Editor Report · Decision Letter 0]

19 Feb 2024

Dear Dr Zhang, 

Thank you for submitting your manuscript entitled "An evolutionarily conserved pathway mediated by neuroparsin in reproductive and behavior plasticity of ants" for consideration as a Research Article by PLOS Biology.

Your manuscript has now been evaluated by the PLOS Biology editorial staff, and I'm writing to let you know that we would like to send your submission out for re-review.

However, before we can send your manuscript back to the reviewers, we need you to complete your submission by providing the metadata that is required for full assessment. To this end, please login to Editorial Manager where you will find the paper in the 'Submissions Needing Revisions' folder on your homepage. Please click 'Revise Submission' from the Action Links and complete all additional questions in the submission questionnaire.

Once your full submission is complete, your paper will undergo a series of checks in preparation for re-review. After your manuscript has passed the checks it will be sent out for review. To provide the metadata for your submission, please Login to Editorial Manager (https://www.editorialmanager.com/pbiology) within two working days, i.e. by Feb 21 2024 11:59PM.

Kind regards,

Roli Roberts

Roland Roberts, PhD

Senior Editor

PLOS Biology

rroberts@plos.org

---

## [Decision Letter · Decision Letter 1]

26 Apr 2024

Dear Dr Zhang,

Thank you for your patience while we considered your revised manuscript "An evolutionarily conserved pathway mediated by neuroparsin in reproductive and behavior plasticity of ants" for publication as a Research Article at PLOS Biology. Your revised study has been evaluated by the PLOS Biology editors, the Academic Editor, and two of the original reviewers.

In light of the reviews, which you will find at the end of this email, we would like to invite you to revise the work to thoroughly address the reviewers' reports.

You'll see that reviewer #1 is now mostly satisfied. However, reviewer #3, while recognising the improvements, still finds the remaining behavioural work to be problematical. S/he also says that code and raw data are missing (we would insist on these anyway, to comply with PLOS data/code availability policy), and that the writing is uneven, being excellent in parts and poor in others.

IMPORTANT: We discussed reviewer #3's more critical comments with the Academic Editor, and sought cross-comments from reviewer #1. Based on these discussions (and similarly negative previous comments from reviewer #4, who was not able to re-review), we will give you one more chance to satisfy the reviewers; if they remain dissatisfied, we will decline to consider your paper further.

Please attend to the following:

a) Please address the remaining comments from the reviewers.

b) Please make all underlying custom code and data available, according to our data policy (see link further down).

c) The Academic Editor noted some similarity between the phylogeny in your Fig 1B and that in Fig 1 of ref 17 (Chandra et al). It looks like it has been re-drawn, so there's no copyright issue, but if ref 17 was indeed the source for the phylogeny then it would be good to cite this paper in the legend.

Given the extent of revision needed, we cannot make a decision about publication until we have seen the revised manuscript and your response to the reviewers' comments. Your revised manuscript is likely to be sent for further evaluation by all or a subset of the reviewers.

**IMPORTANT - SUBMITTING YOUR REVISION**

*Re-submission Checklist*

*Published Peer Review*

*PLOS Data Policy*

*Blot and Gel Data Policy*

Sincerely,

Roli Roberts

Roland Roberts, PhD

Senior Editor

PLOS Biology

rroberts@plos.org

REVIEWERS' COMMENTS:

Reviewer #1:

[identifies himself as Romain Libbrecht]

In the revision of their manuscript, the authors have successfully addressed most of the concerns that I had expressed in my original review. In particular, they have responded to my skepticism related to the behavioral aspects of the study by toning down their interpretation of the behavioral experiments - even removing the behavioral data that I deemed not robust enough because of the experimental design and the method of data collection (see my original review). They have also used a criticism that I made on the possible link between reproduction and behavior - which they had already mentioned in the original manuscript - to reshape their introduction around a possible role of the brain-ovary axis. Then, they have added several ant species to their dataset, which strengthened their comparative analyses. Finally, they improved the writing of the manuscript (note that I still found a small typo L158 and 160: "biori" should be "biroi") and provided the missing details in how they report their statistical analyses.

Overall, I think that these changes greatly improved the manuscript. I thank the authors for the efforts that they have produced in order to address my concerns. I originally recommended the manuscript to be rejected, but I must admit that I was wrong to do so and I am happy that the editors gave the authors the opportunity to resubmit. I find the revised manuscript to be very convincing and, as stated in my original review, I remain very impressed by the fantastic molecular, functional work that is being reported in this study, which I now recommend for publication in Plos Biology.

Romain Libbrecht

Reviewer #3:

I reviewed the previous version of the manuscript. I have now carefully read the other reviews, the author responses, and the revised manuscript. The study topic is very interesting and the results are relatively straightforward and potentially of broad interest. The revised manuscript is improved by some of the additional studies that were done, and some of the changes to the text. However, I find a number of major issues with the revised manuscript, some of which I did not notice in the original version:

1. The behavioral analyses are flawed. 

I agree with Reviewer 1 and Reviewer 4 that the behavioral studies are fundamentally flawed. I did not notice these issues before. As reviewer 4 points out, the statistical analysis needs to explicitly account for the grouping variable by including it as a random factor. The authors responded "Following the reviewer's suggestion, in this revision, we didn't combine the three experiments, instead we presented the results from just one experiment…". This is not an adequate response to the Reviewer 4's comment. As far as I can tell from the Methods, the H. saltator behavioral study used 7 replicate colonies (L313; even though the Methods L681-L688 do not explain replicate colonies, nor L572). In contrast, the behavioral studies in M. pharaonis (L244-248 and L666) use 7-9 individual treated queens (?) placed together in one petri dish and a similar number of untreated queens in a second petri dish. Any random environmental factors that affected one petri dish and not the other would be completely confounded with the treatment effect. This is not a suitable study design. Replicate groups (petri dishes) need to be analyzed and group needs to be explicitly included in the statistical model. 

2. Methodological details, raw data, and scripts are missing. 

In general, I often could not tell exactly what the authors did or how exactly they analyzed their results: as far as I can tell, no scripts or raw data are included, either for the behavioral data or the gene expression data. No mention is made of RNA seq reads being deposited.

3. Writing is inconsistent and difficult to follow in places (especially the Abstract, Introduction, Results, and Methods). 

I had a hard time following the arguments of the Abstract and Introduction (see more details below). In contrast, I found the Discussion to be generally very well written and clear. Overall, the manuscript is inconsistent with the presentation, which in the end greatly detracts from the interesting results and relatively simple and powerful study design (except for the behavior portion, see above).

Additional comments and suggestions:

L40-41 The first sentence of the abstract is stated as if it is a fact, but it's actually a hypothesis. 

"Gyne" is a social insect specific word and should probably be avoided, at least in the Abstract. 

L44 (and L423) what does "remarkable gene regulatory network remodeling" mean? Does this simply refer to transcriptomic differences? I'm used to "remodeling" used with respect to GRNs in terms of their evolution, like this paper: https://www.pnas.org/doi/abs/10.1073/pnas.1305457110?doi=10.1073%2Fpnas.1305457110 . "re-wiring" is used in an evolutionary sense in the Discussion, e.g., L544-560, which fits with my understanding of general usage, but the usage of "remodeling" is confusing in my opinion.

L49 "..mainly targeted on…" is confusing wording. Do you mean that NPA mainly affected..? You showed that injection of NPA peptide affected 5 genes in the brain and 1051 genes in ovaries, so I'm further confused by what "mainly targeted on.." means. 

Sometimes "adjusted P < 0.05" is reported and something "adjusted P < 0.10" is reported. Why? The Methods mentions that that differential expression was analyzed with DESeq2. What exact p adjustment was used? These details need to be specified and relevant scripts used for data analysis need to be included so that reviewers and readers can tell exactly what you analyses you did. Similarly, for GO enrichment with clusterProfiler, please specify what analyses you did, include scripts detailed exactly what you did, and explain how you corrected for multiple comparisons.

L664-L667 refers to "figure generation codes" and notebooks, etc., but as far as I can tell, none of the relevant scripts are actually included, so that it would be impossible for anyone to try to replicate the analyses used in this study.

L51-53 I don't understand this sentence. "Co-opted' suggests evolutionarily co-opted, but co-opted from what? What "distantly related species"? To make this sense, I think you would have to explain the function in other insects and then make inferences about the ancestral state. 

L57-59 this is only true for ant species that have colonies initiated by a single queen. 

L96-99 this wording "suggesting the co-option of the same ..GRN..across ant species" suggests multiple independent co-option events. But it seems more likely that this is ancestral to all extant ants. 

L98-L100 "NPA …earliest …suppressed in queen long after insemination". Please clarify what "earliest" means and also what "long after insemination" means. These are critical introductory points, and these statements are repeated later in the ms, but they are currently unclear. 

L108 what does "anti-juvenile hormone" mean?

Throughout, when referring to "queens", e.g., L138 "..always down-regulated in queens compared to gynes", please clarify: do you mean egg-laying queens, or reproductively mature queens, or simply mated queens?

Fig 1. The estimated 95% CI should be shown instead of 1 SEM. 

L182: What does "a very low level of spontaneous ovary activity" mean? How did you measure it?

L188-189 what does "stagnant ovary activity" mean?

L338-342 this was pharaoh ant mated queens or unmated gynes or? Please explain the source of the 5-12 brains and ovaries. Were they all from the same colony or? These details are important for understanding if the correct statistical analysis was used.

L377-378 why is it relevant that the yellow/MRJP family "contain an array of members in honeybee"? 

L393 "..conserved and crucial function"... specify that by "conserved" you mean conserved across the ant species you studied, and putatively across all ants.

---

## [Decision Letter · Decision Letter 2]

15 Jul 2024

Dear Dr Zhang,

Thank you for your patience while we considered your revised manuscript "An evolutionarily conserved pathway mediated by neuroparsin in reproductive and behavior plasticity of ants" for publication as a Research Article at PLOS Biology. This revised version of your manuscript has been evaluated by the PLOS Biology editors, the Academic Editor, and one of the original reviewers.

Based on the review and our Academic Editor's assessment of your revision, we are likely to accept this manuscript for publication, provided you satisfactorily address the remaining points raised by the reviewer and Academic Editor, and the following data and other policy-related requests.

IMPORTANT - please attend to the following:

a) Please address the remaining concern from reviewer #1. The Academic Editor agrees with this concern, and requests that you de-emphasise the behavioural findings throughout. Also address the additional comments from the Academic Editor (at the foot of this email). 

b) Please change the Title to include an active verb and remove mention of behaviour. We suggest: "An evolutionarily conserved pathway mediated by neuroparsin A regulates reproductive plasticity in ants"

c) Please address my Data Policy requests below; specifically, we need you to supply the numerical values underlying Figs 1B, 2CF, 3BCE, 4CF, 5AB, S2BDE, S3ABDE, S4ADE, either as a supplementary data file or as a permanent DOI’d deposition.

d) Please cite the location of the data clearly in all relevant main and supplementary Figure legends, e.g. “The data underlying this Figure can be found in S1 Data” or “The data underlying this Figure can be found in https://zenodo.org/records/XXXXXXXX

e) Please make any custom code available, either as a supplementary file or as part of your data deposition. We think that this may already be supplied as part of your supplementary files, but please can you confirm?

We expect to receive your revised manuscript within two weeks. 

*Published Peer Review History*

*Press*

Sincerely,

Roli Roberts

Roland Roberts, PhD

Senior Editor

rroberts@plos.org

PLOS Biology

DATA POLICY:

Regardless of the method selected, please ensure that you provide the individual numerical values that underlie the summary data displayed in the following figure panels as they are essential for readers to assess your analysis and to reproduce it: Figs 1B, 2CF, 3BCE, 4CF, 5AB, S2BDE, S3ABDE, S4ADE. NOTE: the numerical data provided should include all replicates AND the way in which the plotted mean and errors were derived (it should not present only the mean/average values).

CODE POLICY

Per journal policy, if you have generated any custom code during the course of this investigation, please make it available without restrictions. Please ensure that the code is sufficiently well documented and reusable, and that your Data Statement in the Editorial Manager submission system accurately describes where your code can be found. [IF APPLICABLE: As the code that you have generated to XXX is important to support the conclusions of your manuscript, its deposition is required for acceptance.]

DATA NOT SHOWN?

REVIEWER'S COMMENTS:

Reviewer #1:

[identifies himself as Romain Libbrecht]

This is the second revision of this manuscript and my assessment remains similar to the last version. I am very impressed by the fantastic molecular, functional work that is being reported and I am very convinced by the data supporting a role of NPA in ant queen reproduction. However, I am still concerned that the behavioral work is not on par with the molecular work, in particular the grouping of queens together in the same container for behavioral analyses and the interdependency between data points that this experimental design produces. Maybe the authors could consider downplaying the behavioral results even more (currently it is still emphasized in the title, abstract, etc).

COMMENTS FROM THE ACADEMIC EDITOR:

I agree with Reviewer 1. Let's have the authors remove the words "and behavior" from the title and de-emphasize the behavioral importance from the rest of the manuscript.

I noticed a few other things that the authors should address as well.

Lines 402 to 406 read like discussion to me. They should consider rewording this or moving it.

The authors should de-emphasize the behavioral results and possibly discuss the limitations of their behavioral experiments in the discussion section.

For Figure 1B I still don’t see the citation to the phylogeny in the figure legend. I agree that they should cite Borowiec. They also don’t mention the phylogeny at all in the legend and they don’t say where the data comes from and what the circles mean. I think this should be described better.

---

## [Editor Report · Decision Letter 3]

23 Jul 2024

Dear Dr Zhang,

Thank you for the submission of your revised Research Article "An evolutionarily conserved pathway mediated by neuroparsin-A regulates reproductive plasticity in ants" for publication in PLOS Biology. On behalf of my colleagues and the Academic Editor, Ingrid Fetter-Pruneda, I'm pleased to say that we can in principle accept your manuscript for publication, provided you address any remaining formatting and reporting issues. These will be detailed in an email you should receive within 2-3 business days from our colleagues in the journal operations team; no action is required from you until then. Please note that we will not be able to formally accept your manuscript and schedule it for publication until you have completed any requested changes.

Sincerely, 

Roli Roberts

Senior Editor

PLOS Biology

rroberts@plos.org